# Comparative Genomic Analysis of Virulent *Vibrio* (*Listonella*) *anguillarum* Serotypes Revealed Genetic Diversity and Genomic Signatures in the O-Antigen Biosynthesis Gene Cluster

**DOI:** 10.3390/microorganisms11030792

**Published:** 2023-03-20

**Authors:** Vimbai Irene Machimbirike, Ignacio Vasquez, Trung Cao, Joy Chukwu-Osazuwa, Oluwatoyin Onireti, Cristopher Segovia, Pongsak Khunrae, Triwit Rattanarojpong, Marije Booman, Simon Jones, Manuel Soto-Davila, Brian Dixon, Javier Santander

**Affiliations:** 1Marine Microbial Pathogenesis and Vaccinology Laboratory, Department of Ocean Sciences, Memorial University of Newfoundland, St. John’s, NL A1C 5S7, Canada; imachimbirik@mun.ca (V.I.M.); ivasquezsoli@mun.ca (I.V.); ttcao@mun.ca (T.C.); jchukwuosazu@mun.ca (J.C.-O.); obonireti@mun.ca (O.O.); cwsegovia@mun.ca (C.S.); 2Department of Microbiology, Faculty of Science, King Mongkut’s University of Technology Thonburi (KMUTT), Bangkok 10140, Thailand; pongsak.khu@kmutt.ac.th (P.K.); triwit.rat@kmutt.ac.th (T.R.); 3The Center for Aquaculture Technologies Canada (CATC), Souris, PE C0A 2B0, Canada; mbooman@aquatechcenter.com; 4Fisheries and Oceans Canada, Pacific Biological Station, Nanaimo, BC V9T 6N7, Canada; simon.jones@dfo-mpo.gc.ca; 5Department of Biology, University of Waterloo, Waterloo, ON N2L 3G1, Canada; msotodavila@uwaterloo.ca (M.S.-D.); bdixon@uwaterloo.ca (B.D.)

**Keywords:** whole genome sequencing, *Vibrio anguillarum*, winter steelhead trout, biochemical identification, vibriosis, lumpfish, pathogenicity

## Abstract

*Vibrio anguillarum* is the most frequent pathogen affecting fish worldwide. The only known virulent strains of *V. anguillarum* are serotypes O1, O2, and O3. Genetic differences between the serotypes that could shed insight on the evolution and serotype differences of this marine pathogen are unknown. Here, we fully sequenced and characterized a strain of *V. anguillarum* O1 (J382) isolated from winter steelhead trout (*Oncorhynchus mykiss irideus*) in British Columbia, Canada. Koch’s postulates using the O1 strain were replicated in naïve lumpfish (*Cyclopterus lumpus*) and compared to O2. Phenotypic and genotypic comparisons were conducted for serotypes O1, O2, and O3, using biochemical tests and bioinformatic tools, respectively. The genome of *V. anguillarum* O1 (J382) contains two chromosomes (3.13 Mb and 1.03 Mb) and two typical pJM1-like plasmids (65,573 and 76,959 bp). Furthermore, *V. anguillarum* O1 (J382) displayed resistance to colistin sulphate, which differs from serotype O2 and could be attributed to the presence of the *ugd* gene. Comparative genomic analysis, among the serotypes, showed that intra-species evolution is driven by insertion sequences, bacteriophages, and a different repertoire of putative ncRNAs. Genetic heterogeneity in the O-antigen biosynthesis gene cluster is characterized by the absence or the presence of unique genes, which could result in differences in the immune evasion mechanisms employed by the respective serotypes. This study contributes to understanding the genetic differences among *V. anguillarum* serovars and their evolution.

## 1. Introduction

Reports of vibriosis in freshwater, brackish water, and marine aquatic animals caused by the Gram-negative bacterium, commonly known as *Vibrio anguillarum* (*Listonella anguillarum*), have been recorded as early as 1893 and continue today [1,2]. This pathogen is important because it causes high morbidity and mortality in aquaculture, resulting in substantial economic losses to the industry. Losses have been reported in fish species of economic importance, such as Atlantic salmon (*Salmo salar*) [3], rainbow trout (*Oncorhynchus mykiss* (Walbaum)), ayu (*Plecoglossiis altivelis*), and eel (*Anguilla japonica*) [4]. Cleaner fish species, such as lumpfish (*Cyclopterus lumpus*), have also been reported to be susceptible to *V. anguillarum* [5].

*V. anguillarum* is composed of 23 O-serotypes, but typically only serotypes O1, O2, and O3 are associated with vibriosis in aquatic animals, while other serotypes are non-pathogenic environmental isolates [6]. Biochemically, *V. anguillarum* is halophilic, catalase- and oxidase-positive, lysine and ornithine decarboxylase-negative, ferments D-glucose without gas production, reduces nitrate to nitrite, is sensitive to the vibriostatic agent 0/129, and cannot ferment inositol or rhamnose [7]. The phenotypic characterization includes motility, growth below 30 °C, non-spore-forming, and facultatively anaerobic metabolism [1,6,8]. Additionally, the extracellular enzyme activities for caseinase, lipase, phospholipase, hemolysin, gelatinase, and siderophore synthesis were recorded for most serotypes, O1, O2, and O3, with a few exceptions [9]. On the genomic level, this bacterium contains two chromosomes, which are around 3.0 and 1.2 Mbp in size and with GC content of 43–46% [10]. Some serotypes O1 and O2 isolates have also been reported to contain virulence plasmids that are 65–67 kbp and are similar to plasmid pJM1 that contains the genes encoding a siderophore-dependent iron-sequestering system [11,12], and at times to contain a smaller plasmid around 12 kbp [5].

Serotyping is a technique used to detect the lipopolysaccharide (LPS) O-antigen by slide agglutination, and in the case of the *V. anguillarum* [13] assessment of the diversity of the O-antigen, separates members of a species into distinct serotypes [14]. The O-antigen is an outer membrane component of Gram-negative bacteria that plays an important role in modulating the immune response of the host species, as well as acting as a bacteriophage receptor. The O-antigen is an oligosaccharide that is composed of between two and six sugar residues. Variations among the serotypes in a species are brought about by the order, nature, and linkage of the sugar components within the oligosaccharide [15]. To date, the structure of the O-antigen from *V. anguillarum* serotypes has been characterized [14,16], but information on the O-antigen gene clusters of the three major serotypes causing an economic impact on aquaculture (serotypes O1, O2, and O3) has yet to be reported.

In the Pacific Northwest, the steelhead trout (*Oncorhynchus mykiss irideus*) is an important fish species for the local economy, culture, and ecosystems [17]. Steelhead trout are affected by several infectious agents like hematopoietic necrosis virus (IHNV) [18], and co-infections of IHNV and *Flavobacterium psychrophilum* [19]. Other diseases reported in steelhead trout include parasitic black spot disease [20] and enteric red mouth disease (ERM) caused by *Yersinia ruckeri* [21]. Currently, there have been no reports of *V. anguillarum* infection in steelhead trout, although infections have been reported in closely related sea-reared rainbow trout [22].

In 1999, a strain of *V. anguillarum* O1, designated here as J382, was isolated from a winter steelhead trout in the Little Campbell River, British Columbia, Canada [23] and used in the present study. This study, therefore, aimed to characterize this strain and describe the characteristics of the complete genome. Phenotypic, biochemical, and genomic characterization confirmed that the *V. anguillarum* J382 strain is serotype O1, closely related to other *V. anguillarum* O1 strains. Virulence was determined in lumpfish. Comparative genomics of *V. anguillarum* serotypes O1, O2, and O3, revealed genetic diversity in the O-antigen biosynthesis gene cluster.

## 2. Materials and Methods

### 2.1. Bacterial Culture Conditions

*V. anguillarum* J382 was isolated from a diseased winter steelhead trout kidney sample in 1999, at a hatchery on the Little Campbell River, British Columbia, Canada [24]. Researchers at the Pacific Biological Station (PBS, Fisheries and Oceans Canada, Nanaimo, British Columbia) sent the *V. anguillarum* strain to our lab for characterization in 2022 (BioSample: SAMN25088161). The strain was identified as serotype O1, using serotyping and a species-specific PCR, using the *rpoS* gene [25]. The bacterium was revived from cryopreservation and routinely grown on trypticase soy agar (TSA; Difco, Franklin Lakes, NJ, USA), supplemented with 2% NaCl and 1.5% bacto agar (Difco, Franklin Lakes, NJ, USA), and incubated for 48 h at 15 °C [5]. The routine cultures produced following the standard procedures, as described above, were maintained in 3 mL of trypticase soy broth (TSB; Difco, Franklin Lakes, NJ, USA), and supplemented with 2% NaCl at 15 °C for 18–24 h until the mid-logarithmic phase was reached (optical density (OD_600_) ≈ 0.7 (~4.1 × 10^8^ CFU/mL)). When required, TSB was supplemented with 100 µM 2,2-dipyridyl (iron-limited condition) and 100 µM of FeCl_3_ (iron-rich condition). Luria-Bertani broth (10 g tryptone, 10 g NaCl; 5 g yeast extract; double distilled H_2_O up to 1 L) was modified to 0%, 0.5%, or 2% NaCl to investigate the NaCl requirements. For investigating the production of siderophores, chrome azurol S (CAS) plates were used [25]. The bacterial hemolytic activity was evaluated using TSA plates supplemented with 5% sheep blood.

### 2.2. Phenotypic Characterization

#### 2.2.1. Physiological Characterization of *V. anguillarum* J382

The bacterial growth was determined under different temperatures, NaCl concentrations, and sugar requirements. For optimum temperature requirement analysis, the microorganism was grown on 2% NaCl TSA at 4, 15, 28, and 37 °C for 48 h. To evaluate the NaCl requirements, the microorganism was grown in 3 mL of LB broth supplemented with 0%, 0.5%, 1.5%, and 2% NaCl and incubated at 15 °C for 24 h. The oxidation/fermentation (OF) of semi-solid media (5.0 g NaCl; 0.3 g di-potassium phosphate; 2.0 g peptone; 0.03 g bromothymol blue; 3.0 g agar; 10 g glucose; double distilled H_2_O up to 1 L) supplemented with 1% of individual sugars (glucose, arabinose, glycogen, maltose, lactose) was used to determine the sugar utilization. The hemolytic activity and motility were determined by standard methods [3].

#### 2.2.2. Siderophore Detection

The secretion of siderophores was tested using CAS plates by previously described assays [5] with minor modification. Briefly, *V. anguillarum* J382 was first cultured in 3 mL 2% NaCl TSB, as described above, at 15 °C for 24 h until the mid-log phase was reached. Subsequently, 30 µL of the inoculum was grown at 15 °C for 24 h, with shaking in 2% NaCl TSB (bacterial control) and the media supplemented with either 100 µM of FeCl_3_ (iron-rich condition), or 100 µM 2,2-dipyridyl (iron-limited condition). After incubation, the cells were pelleted at 4200× *g* at 15 °C for 10 min, washed and resuspended in 1 mL of PBS. The siderophore secretion was analyzed by inoculating 10 µL of the resuspended cells onto the CAS agar plates and incubating for 48–72 h at 15 °C. The bacterial colonies were visually checked for the presence of a yellow-orange halo around them, as an indication of the secretion of siderophores.

#### 2.2.3. Biochemical, Enzymatic, and Physiological Characterization

The biochemical and enzymatic profiles were performed using API 20 NE, API 20 E, and API ZYM strips, according to the manufacturer’s instructions (bioMerieux, Marcy-l’Étoile, France). After incubating the strips with *V. anguillarum* J382 at 15 °C for 48 h, the results were analyzed using apiweb software (bioMerieux). Tests for catalase and oxidase methods were carried out using standard methods [3].

#### 2.2.4. Antibiogram Assays and Vibriostatic Test

The susceptibility and resistance of the bacterial isolate to tetracycline (TET; 30 µg), oxytetracycline (OTC; 30 µg), ampicillin (10 µg), sulfamethoxazole (STX; 25 µg), chloramphenicol (CHL; 30 µg), colistin sulphate (10 mg/mL), oxalinic acid (OXA; 2 µg), and vibriostatic agent 0/129 (O129; 150 µg) was determined using standard methods [3,26]. The choice of the antibiotics tested was based on a previous antibiotic resistance study of *V. anguillarum* serovars [27].

### 2.3. Experimental Fish

Specific pathogen-free lumpfish were produced and reared at the Dr. Joe Brown Aquatic Research Building (JBARB), located at the Memorial University of Newfoundland, following approved animal ethics protocols (#18-1-JS and #18-03-JS). The fish were acclimated in 500 L tanks under optimal conditions (8–10 °C, 95–110% air saturation, ambient photoperiod, and UV-treated, filtered, flow-through seawater). The fish were fed with a commercial feed (Skretting Europa; 55% crude protein, 15% crude fat, 3% calcium, 2% phosphorus, 1.5% crude fiber, 1% sodium, 5000 IU/kg vitamin A, 3000 IU/kg vitamin D, 200 IU/kg vitamin E), administered at a rate of 0.5% of body weight/day, 3 times per day.

### 2.4. Bacterial Infection and Tissue Sampling

Infection assays of *V. anguillarum* J382 were carried out in lumpfish of 50–70 g at the federal certified AQ3 biocontainment zone in the Cold-Ocean Deep-Sea Research Facility (CDRF). Three groups of experimental fish were set up in 3 tanks (*n* = 45 per tank) of 500 L and were acclimatized for 2 weeks under optimal conditions (8–10 °C, 95–110% air saturation, ambient photoperiod, and UV-treated, filtered, flow-through seawater). The *V. anguillarum* J382 was culture, as described (Section 2.1), until the mid-logarithmic phase (optical density (OD_600_) ≈ 0.7 (~4.1 × 10^8^ CFU/mL)). The cells were harvested by centrifugation at 4200× *g* for 10 min at room temperature and washed three times with phosphate-buffered saline (PBS; 136 mM NaCl, 2.7 mM KCl, 10.1 mM Na_2_HPO_4_, 1.5 mM KH_2_PO_4_ (pH 7.2)). After washing, the cells were diluted to ~1.0 × 10^6^ CFU/mL and 1.0 × 10^7^ CFU/mL for bath infection. Quantification of the bacterial was carried out using the plate count method. The lumpfish in tanks 1 and 2 were bath-infected for 30 min with freshly prepared 10^6^ CFU/mL and 10^7^ CFU/mL bacterial inoculum of *V. anguillarum* J382, respectively, while the lumpfish in tank 3 were used as the non-infected control. Mortality was monitored for 15 d and a one-way ANOVA statistical analysis was carried out using GraphPad Prism 9.0.0 (121) software. At 5 days post-infection (dpi), 6 fish were sampled from each tank, euthanized with 400 mg/L of MS222, and the liver, head kidney, and spleen were aseptically removed and placed in sterile homogenized bags (Nasco, Whirl-Pak^®^, Fort Atkinson, WI, USA). The weight of each organ was taken, and the samples were homogenized and adjusted to a final volume of 1 mL (weight: volume) using sterile PBS. The tissue mixtures were serially diluted in sterile PBS (1:10) and plated onto 2% NaCl TSA at 15 °C for 48 h. Bacterial identification was conducted using the API system, according to the manufacturer’s instructions, and bacterial count per gram to tissue was determined as below [28]:$$CFU\times gram-1=\frac{CFU\times \left(\frac{1}{Dilution\;Factor}\right)\times 10}{Tissue\;weight\;\left(grams\right)}$$

### 2.5. Genomic DNA Preparation and Whole Genome Sequencing

*V. anguillarum* J382 was grown until the mid-log phase, as described above. The bacterial cells were then pelleted by centrifugation at 4200× *g* for 10 min at room temperature and washed with sterile PBS 3 times. A commercial DNA extraction kit (Wizard High-Molecular-Weight DNA Extraction Kit; Promega, Madison, WI, USA) was used to extract the genomic DNA, according to the manufacturer’s protocol. The purity and integrity of the extracted DNA was analyzed by spectrophotometry (Genova Nano Spectrophotometer, Jenway; Staffordshire, UK) and on 0.8% agarose gel using gel electrophoresis, respectively. High-quality DNA samples were sent for library preparation and whole genome sequencing using 2 platforms, namely, PacBio SMRT cell and Illumina MiSeq PE250, at Genome Quebec (Montreal, QC, Canada). The whole genome sequence was submitted to the National Center for Biotechnology Information (NCBI) database.

### 2.6. Genomes Used in This Study

A total of 13 complete genomes of *V. anguillarum* strains from serotypes O1, O2, and O3, isolated from diverse environments, as well as *V. fluvialis* ATCC 33809, *V. parahaemolyticus* R14, *Vibrio cholerae* RFB05, *Vibrio splendidus* BST398, *Vibrio campbellii* ATCC 25920, CAIM 519T and *Photobacterium damselae* subsp. piscicida 91-197, were obtained from the NCBI (Table 1).

### 2.7. Data Preprocessing, Genome Assembly, Mapping and Annotation

The PacBio reads were assembled at Genome Quebec using the Hierarchical Genome Assembly Process (HGAP) assembler v.4 [29] with 30X coverage cutoff [28,30]. Confirmation of the assembly was carried using the CLC Genomics Workbench (CGWB) v.20.0.4 (Qiagen, Hilden, Germany). The Illumina raw reads were trimmed, and low-quality reads were filtered out using the CGWB and the reads quality was analyzed by FastQC v12 [31]. The PacBio long reads were assembled into circular contigs using the CGWB. The PacBio assembled genome was polished using the trimmed high-quality Illumina reads and the CGWB de novo tool in the genome finishing module tools with default settings. The plasmids were assembled using the Illumina reads that did not align with the *V. anguillarum* J382 chromosomes, following the above-mentioned assembly protocol. The assembled chromosomes and plasmids were annotated using the Rapid Annotation Subsystem Technology pipeline (RAST) [32] and comprehensive genome analysis implemented in PATRIC v3.6.12 [33]. The assembled chromosomes and plasmids were re-annotated by the NCBI prokaryotic genome annotation pipeline (PGAP). The obtained 2 chromosomes and plasmids were further mapped using the web server CGView Server^BETA^ with Proksee annotation (https://sites.ualberta.ca/~stothard/ (accessed on 13 March 2023)).

### 2.8. Plasmid Analysis

The number of plasmids in *V. anguillarum* J382 was determined using alkaline extraction, described elsewhere [34]. The extracted plasmids were detected on 0.5% agarose gel at 40 V overnight. The gels were stained with ethidium bromide and viewed using the Invitrogen iBright CL1500 Imaging System (Thermo Fisher Scientific, Waltham, MA, USA).

### 2.9. Comparative Genomic Analysis, Synteny and Phylogeny Analysis

The CGWB was used to compare the genomes listed in Table 1. Chromosomes 1 and 2 were analyzed separately. Whole genome multiple alignment was performed using the whole genome analysis tool with default settings (minimum initial seed length = 15 (allow mismatches in seeds), minimum alignment block length = 100 (rearrange contigs)). The average nucleotide identity comparison was conducted using the default parameters: minimum similarity fraction = 0.8, minimum length fraction = 0.8. Comparative heat maps were constructed following the default parameters of the distance measure of 1 Pearson correlation and complete linkage criteria. For synteny studies, whole genome dot plots were constructed between *V. anguillarum* J382 and representative genomes in serotypes O1 (*V. anguillarum* 87-9-116), O2 (*V. anguillarum* J360), and O3 (*V. anguillarum* CNEVA NB11008), using the parameters: minimum initial seed length = 15 and allow mismatches in the seeds (default parameters). The separate whole genome multiple alignment files for chromosomes 1 and 2 created using the CLC Bio Workbench were exported as FASTA files, used to construct phylogenetic trees in MEGA 11 [35] using the neighbor-joining statistical method [36] with 1000 replicate bootstraps. The strain *P. damselae* subsp. *Piscicida* 91-197 was used as the out group to root the tree.

### 2.10. Determination of O-Antigen Biosynthesis Genes

The O-antigen biosynthesis genes, their corresponding pathways, and the sugars they encode were determined using bioinformatic tools. The KEGG platform [29] was used to annotate and identify the O-antigen biosynthesis genes in *V. anguillarum* J382 and the representative genomes in the serotypes O1 (*V. anguillarum* 87-9-116), O2 (*V. anguillarum* J360), and O3 (*V. anguillarum* CNEVA NB11008). Firstly, whole genome amino acid sequences in FASTA format were downloaded from either RAST for *V. anguillarum* J382 or from the NCBI for the remaining genomes, and KO identifiers/K numbers (assignment of orthologous genes based on sequence similarity) were assigned to the genes. Two platforms implemented in KEGG were used for the K number assignment, namely BlastKOALA [29] that uses BLASTP sequence similarity search (parameters: taxonomy group of the genome, Bacteria and KEGG GENES database search file, genus_prokaryotes) and GhostKOALA [37], which performs automatic KO assignment by GHOSTX sequence similarity search (parameters: KEGG GENES database search file, genus_prokaryotes). The resulting files were uploaded into KEGG Mapper, Reconstruct [38] for O-antigen biosynthesis pathways identification in the genomes, including the resulting oligosaccharides. The predicted O-antigen oligosaccharides in each serotype were manually presented as a Venn diagram. Comparative analysis and visualization of the O-antigen biosynthesis genes, among the three serotypes, was performed using ggplot2 [39] implemented in R statistical software (v4.1.2; R Core Team 2021). The LPS profiles were evaluated by SDS-PAGE and visualized by silver staining [40,41,42].

### 2.11. Prediction of Non-Coding RNAs

The non-coding RNAs (ncRNAs) of V. *anguillarum* J382 were predicted and computed, as described earlier [43]. The pipeline included ncRNA identification, annotation, and taxonomic RNA assignation, based on the secondary structure using StructRNAfinder [44,45,46,47,48]. The ncRNAs in the *V. anguillarum* J360 genome [4] were retrieved (29 October 2022) and used for comparison with serotype O1. A comparison between the filtered ncRNAs of *V. anguillarum* J382 (serotype O1) and *V. anguillarum* J360 (serotype O2) was plotted using jvenn [49]. The functions of the predicted ncRNAs were explored using the Rfam database.

### 2.12. Statistical Analysis

Statistical analysis of the mortality between the bath challenged groups and the control groups was conducted using a one-way ANOVA in GraphPad Prism 9.0.0 (121).

## 3. Results

### 3.1. Phenotypic Characterization of V. anguillarum J382

The phenotypic characterization of *V. anguillarum* J382 shown in Table 2 and Appendix A, revealed that this strain exhibited the characteristics of marine *Vibrio* spp. [50,51]. These included halophilic metabolism; positive reactions for indole, catalase, and oxidase; the fermentation of D-glucose, mannitol, sorbitol, and arabinose; the reduction of nitrate to nitrite; and the hydrolysis of L-arginine. *V. anguillarum* J382 was sensitive to the vibriostatic agent 0129, and the antibiotics chloramphenicol, oxytetracycline, tetracycline, sulfamethoxazole, and oxalinic acid, and resistant to ampicillin, similar to other members in the genus [50]. Interestingly, *V. anguillarum* J382 was resistant to colistin sulphate. The siderophore-dependent iron acquisition was also analyzed. *V. anguillarum* J382 was able to synthesize siderophores under iron-limited conditions (2% NaCl TSB supplemented with 100 µM 2,2-dipyridyl), whilst sideorophore synthesis was not observed under iron-rich conditions (2% NaCl TSB supplemented with 100 µM FeCl_3_). The hemolysin activity of *V. anguillarum* J382 on sheep blood agar was temperature dependent. Hemolysis was observed after incubation at 28 °C, while there was no hemolysis at 15 °C.

### 3.2. Clinical Signs and Fish Mortality

The lumpfish bath-infected with *V. anguillarum* J382 exhibited clinical signs that are consistent with vibriosis, including skin discoloration, hemorrhage on the fins and fish body, lesions around the mouth, reddened anus, bloody eyes, and exophthalmia (Figure 1a–g). Mortality in the two doses evaluated, started at 5 days post-infection (dpi) and all the fish were dead by 7 and 9 dpi after the high (10^7^ CFU/mL) and lower dose (10^6^ CFU/dosemL), respectively (Figure 1h). There was a significant difference in mortality between the infected groups and the control group (*p* = 0.0153).

### 3.3. Tissue Colonization of V. anguillarum J382 Infected Lumpfish

Tissues from all the organs (heart, spleen, head kidney, and liver) isolated from the lumpfish infected with 10^6^ CFU/mL or 10^7^ CFU/mL doses of *V. anguillarum* J382 were heavily colonized with bacteria. The bacterial loads ranged from >10^7^ CFU/g tissue in the heart to >10^10^ CFU/g tissue in the other tissue organs (Figure 1i,j). The highest bacterial load (up to 10^12^ CFU/g tissue) was recorded in the spleen for both the 10^6^ CFU/mL and 10^7^ CFU/mL doses. The bacteria isolated from the colonized fish organs were characterized using the API system (Appendix A) and the phenotypic characterization confirmed the identity of the recovered bacteria as *V. anguillarum*.

### 3.4. Genome Sequencing and Characterization

The complete genome of *V. anguillarum* J382 consists of two chromosomes and two plasmids (accession numbers CP091185 for chromosome 1, CP091186 for chromosome 2, CP091187 for plasmid pVA_O1_1, and CP091188 for plasmid pVA_O1_2). The sizes of the two chromosomes are 3,133,133 bp and 1,038,699 bp (Appendix A), with GC content of 44.64% and 44.04%. Chromosome 1 has 300 subsystems, 2914 protein-coding sequences (CDS), 95 tRNA, and 28 rRNA sequences (Figure 2a). On the other hand, chromosome 2 has 58 subsystems, 1007 CDS, and 4 tRNA sequences (Figure 2b). Appendix A shows the genes associated with pathogenesis and environmental adaption that were identified in chromosomes 1 and 2. Chromosome 1 contains most of the genes related to virulence, whose functions are related to iron acquisition and metabolism, motility, and chemotaxis, siderophore biosynthesis and transport, type II and IV secretion systems and toxins, and superantigens. Both chromosomes contain the genes related to cell wall and capsule formation, invasion and intracellular resistance, resistance to toxic compounds, antibiotic resistance, and hemolysins.

Two similar plasmids (pVA_O1_1 and pVA_O1_2) were identified in *V. anguillarum* J382, as shown in Figure 3a,b. Summarized details of the two plasmids are shown in Appendix A. The plasmid pVA_O1_1 is 65,573 bp in length, whilst the plasmid pVA_O1_2 is 76,959 bp in length, with similar GC content of 42.59% and 42.6%, respectively. Both plasmids have two subsystems, and 69 CDS for plasmid pVA_O1_1 and 106 CDS for plasmid pVA_O1_2. The genes contained in both plasmids are the same except that plasmid pVA_O1_2 contains multiple copies of genes, such as 4′-phosphopantetheinyl transferase (*entD*), ferric vibriobactin, enterobactin transport system, ATP-binding protein (*viuG*), isochorismatase (*entB*), isochorismate synthase (*entC*), 2,3-dihydroxybenzoate-AMP ligase (*entE*), histidine decarboxylase (*hdc*), and genes that encode for the efflux ABC transporter, permease/ATP-binding protein, transport ATP-binding protein CydCD, and non-ribosomal peptide synthetase modules, siderophore biosynthesis (Appendix A). The genes associated with virulence and adaptation identified in both plasmids are related to iron acquisition and metabolism, siderophore biosynthesis and transport, and toxins and superantigens (Appendix A). Verification of the presence of the plasmids in *V. anguillarum* J382 revealed a single DNA band around 23 Mda (Figure 3c), which could be due to the small size difference between the plasmids. A blastn search in the NCBI database revealed that both plasmids, pVA_O1_1 and pVA_O1_2, are 100% and 99.98% identical to plasmid pJM1, respectively (Appendix A).

### 3.5. Comparative Genomics

#### 3.5.1. Average Nucleotide Identities (ANI) and Phylogeny

The average nucleotide identities between the *V. anguillarum* J382 chromosomes 1 and 2, and chromosomes from other *V. anguillarum* strains, revealed high ANI values (99–100%), especially with serotype O1 strains. The ANI values and GC% among the *V. anguillarum* strains were conserved regardless of the fish host and spatial distribution (Table 1). Both chromosomes 1 and 2 of *V. anguillarum* J382 are very similar to chromosomes 1 and 2 from the serotype O1 strains *V. anguillarum* 775, *V. anguillarum* M3, *V. anguillarum* 87-9-116, *V. anguillarum* 425, and *V. anguillarum* ATCC-68554 (Figure 4a,b).

The phylogenetic relationships provided insights on the closest neighbors of the *V. anguillarum* J382 chromosomes 1 and 2. The neighbor-joining method revealed that chromosome 1 from whole genomes of *V. anguillarum* are clustered into four groups (Figure 4c). Cluster I consists of serotype O1, cluster II consists of serotype O2, cluster III consists of serotype O3, and cluster IV consists of a strain of unknown serotype. Phylogenetic analysis of *V. anguillarum* J382 chromosome 2 revealed five clusters (Figure 4d). Clusters I and III consisted of chromosomes 2 from serotype O1, whilst clusters II, IV, and V consisted of serotypes O2, O3, and unknown/other, respectively. Surprisingly, both chromosomes 1 and 2 of *V. anguillarum* VIB43, a strain that was earlier reported to be serotype O1, clustered together with serotype O2 strains, *V. anguillarum* J360 and VIB12. Both phylogenetic trees revealed that in the *Vibrio* genus, *V. anguillarum* is closely related to *V. cholerae.*

#### 3.5.2. Synteny Analysis

Comparisons between the chromosomes of *V. anguillarum* J382 with those of con-specific isolates were performed to study synteny. Dot plot visualization of the serotype O1 chromosome 1 (*V. anguillarum* J382 vs. *V. anguillarum* 87-9-116) showed high similarity (Figure 5a), although there was genomic rearrangement of the locally collinear blocks (Appendix A). The visualization of *V. anguillarum* J382 chromosome 1 against *V. anguillarum* J360 chromosome 1 (serotype O2) (Figure 5b), and *V. anguillarum* J382 chromosome 1 against *V. anguillarum* CNEVA NB 11008 chromosome 1 (serotype O3) (Figure 5c), revealed genomic gaps, inversions, and orthologs. This was supported by the locally collinear blocks arrangements that also showed genomic rearrangements (Appendix A). Similar results were observed when synteny was studied for *V. anguillarum* J382 chromosome 2 (Figure 5d–f and Appendix A).

#### 3.5.3. Comparison of O-Antigen Biosynthesis Gene Cluster in *V. anguillarum* Serotypes

Diversity of the O-antigen biosynthesis gene cluster was observed among the *V. anguillarum* serotypes O1, O2, and O3 (Figure 6). The representative serotype of the O1 genomes, *V. anguillarum* 87-9-116 and J382, both contain 24 genes related to O-antigen biosynthesis, of which 22 genes are found in chromosome 1, and 11 genes and are unique to serotype O1 (Appendix A). These genes in serotype O1 are involved in nine biosynthesis pathways that result in the presence of eight sugars and oligosaccharides in the O-antigen, including CDP-4-dehydro-3,6-dideoxy-D-glucose, UDP-α-d-galacturonic acid (UDP-GalA), dTDP-L-rhamnose, dTDP-4-amino-4,6-dideoxy-D-galactose, GDP-perosamine, UDP-N-acetyl-α-D-mannosaminuronate, UDP-N-acetyl-D-quinovosamine, and 2-acetamido-3-amino-2,3-dideoxy-D-glucuronamide (Figure 7a and Appendix A). On the other hand, representatives of serotype O2 (*V. anguillarum* J360) and serotype O3 (*V. anguillarum* CNEVA NB 11008) contain 13 and 14 O-antigen biosynthesis genes, respectively (Appendix A). In serotype O2, the genes *wbpE*/*wlbC*, *wbpI*/*wlbD*/*wecB*, and *wlbA*/*wbplA* are unique, whereas the gene *wbp*P/*tvi*C is unique to serotype O3. Both, serotype O2 and O3 have genes involved in seven biosynthesis pathways that produce six sugars and oligosaccharides in the O-antigen. Serotype O2 has the putative ability to synthetize UDP-glucose, mannose-1-phosphate, UDP-N-acetyl-α-D-mannosaminuronate, 2,4-diacetamido-2,4,6-trideoxy-u-glucose (N,N′-diacetylbacillosamine)2-acetamido-3-amino-2,3-dideoxy-D-glucuronamide, and 2,3-diacetamido-2,3-dideoxy-d-mannuronic acid (Appendix A). Serotype O3 has the putative ability to synthetize UDP-glucose, dTDP-4-oxo-6-deoxy-D-glucose, mannose-1-phosphate, UDP-N-acetyl-α-D-mannosaminuronate, UDP-N-acetyl-D-quinovosamine, and uridine diphosphate N-acetylgalactosamine (Figure 7a and Appendix A).

The genes *galE*, *manA*, *manB*, *wecB*, *wecC*, *glmU*, *wbp,* and the oligosaccharide UDP-N-acetyl-α-D-mannosaminuronate, are common among the three serotypes (Figure 7a and Appendix A). Interestingly, the *wecA* gene is only found in serotype O1, and *wbp*O is unique to both serotypes O2 and O3 (Appendix A). In the three serotypes studied, two genes (*galE* and *manB*) are found in chromosome 2 and the rest of the studied genes in chromosome 1 (Appendix A).

Only one oligosaccharide is common to O1, O2, and O3 serotypes. One oligosaccharide is also shared between O1 and O2, or between O1 and O3, and two oligosaccharides are shared between O2 and O3 (Figure 7a).

Lipopolysaccharide extracted from isolates, *V. anguillarum* J382 (O1) and *V. anguillarum* J360 (O2) displayed distinct LPS profiles between the two serotypes (Figure 7b). Several LPS bands were missing in serotype O2 in the O-antigen protein region. Also, there were differences in the core/lipid A protein band in the band sizes, as well as the number of bands.

#### 3.5.4. Characteristics of ncRNAs in *V. anguillarum* Serotypes O1 and O2

A total of 122 ncRNA were annotated in *V. anguillarum* J382, of which 89 ncRNAs and 33 ncRNAs were in chromosome 1 and chromosome 2, respectively (Appendix A). There were multiple copies of the ncRNAs CsrB_RF00018, CsrC_RF00084, Thr_leader_RF00506, Leu_leader_RF00512, MicX_RF01808, RsaJ_RF01822, and FsrA_RF02273 in both chromosomes. Multiple copies of the ncRNAs TPP_RF00059, ctRNA_p42d_RF00489, Acido-Lenti-1_RF01687, rimP_RF01770, Phe_leader_RF01859, and BjrC68_RF02353, were found in *V. anguillarum* J382 chromosome 1 only, whilst multiple copies of the ncRNAs Qrr_RF00378, MIR529_RF00908, and ar35_RF02346 were found only in *V. anguillarum* J382 chromosome 2 (Appendix A).

The comparison of annotated ncRNAs between serotype O1 and O2 yielded 47 common ncRNAs, while 38 were predicted only in serotype O1 (J382) and 14 only in serotype O2 (Appendix A). The comparison revealed that there are ncRNAs predicted only in one serotype and absent in another (Figure 8). The ncRNAs that were predicted only in serotype O1 are linked to functions, such as pathogenesis, RNA modification, and the bacterial signal recognition particle RNA (Figure 8a). Both serotypes have ncRNAs that originate from plasmids and viruses, but are not commonly shared (Figure 8a,b). Commonly, ncRNAs linked to functions, such as Hfq binding, protein synthesis, metabolism, virulence, iron regulation, stress response, quorum sensing, riboswitches, ribozymes, and transport, are found in both genomes of the two serotypes (Figure 8c).

## 4. Discussion

Infections with *V. anguillarum* in fish cause significant losses in the marine environment and aquaculture, although vaccines have reduced the risk of vibriosis in many of these cultured species [2]. The repertoire of aquatic animals that are susceptible to *V. anguillarum* includes important species, such as rainbow trout, salmon, turbot, sea bream, sea bass, ayu, cod, and eel [2]; cleaner fish such as lumpfish and wrasse [52]; ornamental fish [53]; and cultured bivalves [54]. *V. anguillarum* infections also have socio-economic impacts on the aquaculture industry, whereby the mortalities of aquatic animal species can reach up to 100%. For instance, annual economic losses due to *V. anguillarum* have been estimated to be between USD 18–30 million in Japan [54]. Herein, we characterized a strain of *V. anguillarum* serotype O1, isolated from winter steelhead trout in 1999. Although *V. anguillarum* is commonly reported in rainbow trout, this strain was isolated from the first reported vibriosis in winter steelhead [22]. The virulence of strain J382 was confirmed in lumpfish, where there was 100% mortality at 9 dpi in lumpfish (Figure 1).

The *V. anguillarum* serotype O1 J382 showed a typical *V. anguillarum* phenotype and genome. It was earlier reported that most *V. anguillarum* serovar O1 exhibited resistance to colistin, but were sensitive to ampicillin and cephalothin, whilst the opposite was observed for most serovar 02 strains [27]. The *V. anguillarum* O1 (J382) strain was particularly resistant to colistin sulphate, which was contrary to the *V. anguillarum* serotype O2 (J360) isolated from lumpfish [5]. Previously, it has been suggested that some intrinsic antimicrobial resistance in the *V. anguillarum* serovar is dependent and the genes responsible are chromosomally encoded [27]. This was supported by the genomic analysis of *V. anguillarum* J382, where all the antibiotic-resistant genes were located on either chromosome 1 or chromosome 2 (Appendix A).

Phenotypically, members of *V. anguillarum* are eurythermal having an optimal growth temperature of between 30–34 °C, with limited growth below 5 °C, but grow rapidly between 25–30 °C [55,56]. *V. anguillarum* have a halophilic metabolism and thrive at NaCl concentrations between 1% and 2%, allowing for survival in seawater for over 50 months. Some strains have a plasmid-mediated iron-sequestering system [2,55]. The characteristics exhibited by *V. anguillarum* J382 were in accordance with those mentioned above. Our study also verified that the virulence of *V. anguillarum* is dually modulated by temperature and iron availability. For instance, hemolysin activity was noted at 28 °C on sheep blood agar, whilst there was limited activity at 15 °C, and the iron-sequestering ability of *V. anguillarum* O1 was demonstrated under iron-limited conditions (Appendix A).

Previous transcriptomic analysis of the *V. anguillarum* serotype O2 revealed that under iron starvation some virulence factors like LPS biosynthesis, heme receptor HuvA, siderophore piscibactin, MARTX toxins, and T6SS2 were upregulated at 15 °C, while hemolysin Vah1, T6SS1, ferrous iron transport, and the vanchrobactin siderophore system were upregulated at 25 °C [56]. Chemotaxis and flagellum-related genes were downregulated under iron starvation and at 15 °C. This iron and temperature dependent behavior of *V. anguillarum* needs to be considered when producing vaccines, as the vaccine might lack some important antigens thereby affecting the protection [56].

The genotypic characterization of *V. anguillarum* J382 revealed the presence of two chromosomes and two pJM1-like plasmids. The presence of two chromosomes is a general characteristic of *Vibrio* spp. that evolved as a survival strategy, allowing the rapid adaptation of the pathogen to different environments, such as seawater and different hosts [2]. The smaller chromosome 2 is thought to have been a megaplasmid that was taken up by an ancestral *Vibrio* spp., since its *par*A gene tends to cluster with homologues from plasmids and not from chromosomes [57], contains an integron region found in plasmids, contains only four tRNA genes, and no rRNA gene is present (Appendix A). Chromosome 2 could have acquired genes from other species, as well as from chromosome 1, since most of its genes are identical copies [57]. On the other hand, the two pJM1-like plasmids found in *V. anguillarum* J382 are characteristic of serotype O1, which is absent in serotype O2 and encode an iron-sequestering system that is siderophore-dependent (siderophore anguibactin) [2]. The iron-sequestering system can efficiently dislocate host protein-bound iron making it available for uptake by the bacteria [54]. The iron-sequestering role is shared between the pJM1-like plasmids and chromosome 1 (Appendix A) in serotype O1 strains. Another iron-uptake system found in O1 and O2 *V. anguillarum* serovars, known as the siderophore vanchrobactin, is independent of the pJM1 plasmid [54].

The average nucleotide identity (ANI), phylogeny, and synteny of *V. anguillarum* J382, with close and distant related species, was carried out. The *V. anguillarum* J382 chromosomes consistently showed similarity to the other serotypes O1 genomes, whilst the identity decreased when compared to serotype O2, O3, and an unspecified serotype strain. Whole genome ANI is a bioinformatic approach corresponding to DNA–DNA hybridization, which is a simple and useful method for the purpose of describing genetic relatedness that can effectively delineate bacterial species based on lineage-specific genes [57]. Evidence of evolution within the *V. anguillarum* species, especially among serotypes O1, O2, and O3, was displayed by the presence of genomic rearrangements, deletions, and inversions, as well as the presence of orthologs. However, the synteny of locally colinear blocks, among the three serotypes, revealed the conservation of functionally related genes. Colinear syntenic blocks are defined as conserved groups of genes across taxa that are consistently encoded in the same locality for gene expression regulation in prokaryotes [58,59]. The whole chromosome phylogenetic analysis of *V. anguillarum* confirmed earlier reports whereby the three serotypes were distinctly separated [5].

The genes encoding for O-antigen biosynthesis arranged in a gene cluster are generally located on the chromosomes and are separated into three groups, namely, nucleotide sugar biosynthesis, sugar transfer, and O-antigen processing [15]. The predicted *V*. *anguillarum* O-antigen gene clusters displayed variations among serotypes O1, O2, and O3. Evidence of inter-species lateral transfer of the O-antigen gene clusters exists and is partly attributed to O-antigen diversity in Gram-negative bacteria [15]. Common in most bacteria, sugars or sugar derivatives that comprise the O-antigen components are transferred from nucleotide sugar precursors [15]. In the present study, nucleotide sugar precursors uridine 5′-diphosphate-glucose (UDP-Glc), uridine diphosphate galactose (UDP-Gal), and uridine diphosphate N-acetylglucosamine (UDP-GlcNAc) that are common among Gram-negative bacteria [60] were also predicted in the three *V. anguillarum* serotypes. On the other hand, the UDP-sugar precursor synthesizing gene, *ugd and wecA* gene [15] were only identified in serotype O1.

The *V. anguillarum* J382 (serotype O1) contains genes for the synthesis of 6-deoxyhexose sugars/dTDP-sugars (L-rhamnose and 4-acetamido-4,6-dideoxy-D-galactose) from glucose-1-phosphate via the dTDP-sugar biosynthesis pathways. The serotype also contains CDP-sugar and GDP-sugar biosynthesis pathways, which produce CDP-4-dehydro-3,6-dideoxy-D-glucose and perosamine, respectively. The presence of the L-rhamnose and 3,6-dideoxyhexose biosynthesis gene clusters in the *V. anguillarum* serotype O1 was previously reported [15,61]. The other sugar biosynthesis pathways present in serotype O1 are the UDP-N-acetylglucosamine based pathways that utilize GlcNAc-1P producing acetyl-α-D-mannosaminuronate, 2-acetamido-3-amino-2,3-dideoxy-D-glucuronamide, and N-Acetyl-D-quinovosamine. N-acetyl-D-quinovosamine and 2-acetamido-3-amino-2,3-dideoxy-D-glucuronamide, which is a precursor of UDP-2,3-diacetamido-2,3-dideoxy-d-mannuronic acid (UDP-ManNAc3NAcA), have only been reported in *Bordetella* spp., and *P. aeruginosa* [15,62], but not in other *Vibrio* spp., which supports the evidence of inter-species lateral gene transfer among Gram-negative bacteria. Furthermore, *V. anguillarum* J382 exhibited resistance to colistin sulphate, which was not observed in serotype O2 [5]. The *ugd* gene that was unique to serotype O1 is known to confer resistance to cationic antimicrobial peptides and antibiotics, such as polymyxin and colistin [40]. This is because the Etk-mediated phosphorylated Ugd plays a role in the synthesis and transfer of 4-amino-4-deoxy-L-arabinose (Ara4N) to lipid A and capsular polysaccharide biosynthesis [39,62,63,64,65]. The *wecA* gene identified only in serotype O1, is common in *Enterobacteriaceae* and is situated outside of the O-antigen gene cluster. Its role is to transfer a sugar phosphate to carrier lipid, undecaprenyl phosphate (UndP) [15]. Moreover, the presence of the *rfbH* and *rfbI* in serotype O1 genes, which are homologs of *wzm* and *wzt* in *V. cholerae* [62,66], indicates that this serotype utilizes the Wzm/Wzt export system pathway for O-antigen translocation and polymerization [15].

The genomic characterization of *V. anguillarum* serotype O2 supported earlier findings of the LPS analysis that was conducted using nuclear magnetic resonance techniques and partial hydrolysis [16]. In this study, *V. anguillarum* J360 used as a serotype O2 representative strain, was shown to include only UDP-sugar biosynthesis pathways for the production of N-acetyl-α-D-mannosaminuronate, 2,4-diacetamido-2,4,6-trideoxy-u-glucose (N,N’-diacetylbacillosamine), 2-acetamido-3-amino-2,3-dideoxy-D-glucuronamide, and 2,3-diacetamido-2,3-dideoxy-d-mannuronic acid. These polysaccharides were earlier identified in serotype O2, together with an N-formyl-L-alanyl group [16]. For serotype O3, one dTDP-sugar biosynthesis and three UDP-sugar biosynthesis pathways were identified by bioinformatic analysis of the representative whole genome (CNEVA NB 11008). The *wbpO*/*tviB* gene unique to serotypes O2 and O3 is a key enzyme in the B-band LPS production of *P. aeruginosa* O6 [67]. This finding may suggest the presence of the LPS B-band in the two serotypes used. However, further LPS characterization experiments with these serotypes are required to elucidate this genetic finding. Furthermore, the absence of the *rfbH* and *rfbI* required for the Wzm/Wzt export system pathway suggests that both serotypes utilize alternative pathways for O-antigen translocation and polymerization. We, however, did not recognize any genes homologous to the genes encoding for Wzx and Wzy utilized by the Wzx/Wzy pathway, or the genes encoding WbbE and WbbF utilized by the synthase pathway in the O-antigen gene clusters [68]. The responsible genes might be located elsewhere on the chromosome as was found in *Salmonella enterica* groups D1 and B, where the *wzy* gene was found outside the O-antigen gene cluster [69].

Bacterial non-coding RNAs (ncRNA) are known to carry out various functions that range from genetic information processing, physiological adaptation, and metabolism [70,71]. The discovery of bacterial ncRNAs has primarily been conducted using computational searching using genomic and metagenomic sequence data [69]. Bacterial ncRNAs were first found within intergenic regions of *E. coli*, with high conservation among closely related species [72]. Herein, we annotated ncRNAs in the serotype O1 *V. anguillarum* J382 genome using computational tools. To check for the conservancy of the ncRNAs, a comparison was carried out between *V. anguillarum* J382 and *V. anguillarum* J360, a serotype O2 strain. A total of 122 ncRNAs were predicted in *V. anguillarum* J382, whose functions were linked to physiological processes, such as virulence, quorum sensing, and stress responses.

The comparison of ncRNAs in both *V. anguillarum* serotypes revealed that several of the common ncRNAs were linked to pathogenesis of the bacteria to the host. These include several Hfq binding ncRNAs, such as Qrr_RF00378. The Hfq chaperone in conjunction with qrr sRNAs in *V. cholerae* and related sRNAs in *V. harveyi*, was shown to destabilize the mRNA encoding the regulators LuxR and HapR in *V. harveyi* and *V. cholerae*, respectively, thereby suppressing quorum sensing [73]. Other ncRNAs involved in pathogenesis include the VrrA (VrrA_RF01456), which in *V. cholerae*, influence intestinal colonization and adhesion by regulating the expression of TcpA and OmpA, respectively, as well as and outer membrane vesicle (OMV) formation [74]. The ncRNAs like CsrB/CsrC (CsrB_RF00018/CsrC_RF00084), Spot 42 (Spot_42_RF00021), and RyhB (RyhB_RF00057) are also involved in the regulation of virulence factors, namely, expression, cytotoxicity, and iron metabolism, respectively, of *Vibrio* species [75]. Despite the limited attention given to bacterial microRNAs (miRNAs), they have been proven to exist in bacterial species, such as *Streptococcus mutans*, *E. coli,* and *Thalassospira* spp. [76], and in this study, miRNAs were also identified in both serotypes. Future research is required to elucidate the role of the miRNAs in bacteria.

## 5. Conclusions

*V. anguillarum* O1 J382, isolated from a natural infection in winter steelhead trout in the Pacific coast, was subjected to phenotypic and genomic characterization. *V. anguillarum* J382 isolate was highly virulent in Atlantic lumpfish, suggesting a broad host fish range. Also, J383 was colistin sulphate resistant, like other *V. anguillarum* O1 strains. Comparative genomics of the O-antigen gene clusters of serotypes O1, O2, and O3 revealed genetic variation in genes related to nucleotide sugar translocation mechanisms, O-antigen polymerization, and composition. Moreover, gene rearrangements between the serotypes were observed. The ANI, phylogeny, and synteny revealed the closest relatives of *V. anguillarum* J382, as well as genetic diversity among the different serotypes showing intra-species evolution. Some of the O-antigen biosynthesis pathways identified were earlier identified in distantly related Gram-negative bacteria, such as *P. aeruginosa*, indicating the inter-species lateral transfer of the O-antigen gene clusters. The ncRNAs were also annotated and they provided insight into the various putative functions they might carry out in *V. anguillarum* under physiological adaptation. This study contributes essential pathogenesis and comparative genomics data on virulent *V. anguillarum* serotypes, critical for further fundamental studies (e.g., gene function) and applications in fish health (e.g., vaccines).

## Figures and Tables

**Figure 1 microorganisms-11-00792-f001:**
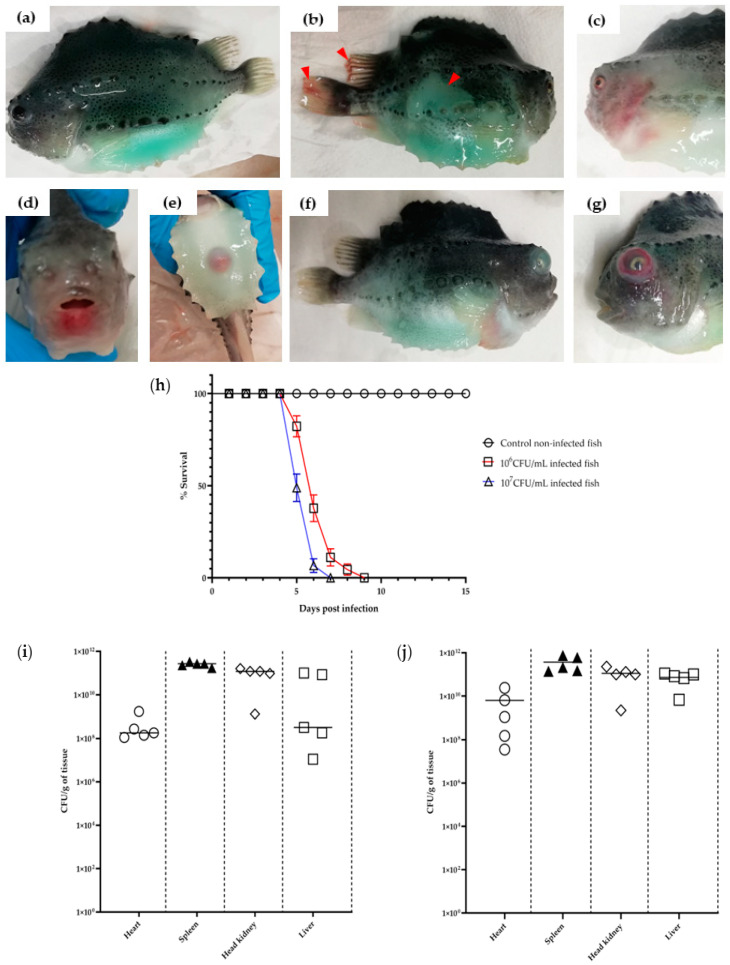
Clinical signs and daily survival rate of fish infected with *V. anguillarum* J382. The lumpfish (weight 50–70 g) were infected with either approximately 10^6^ CFU/mL or 10^7^ CFU/mL of *V. anguillarum* J382. (**a**) Shows the uninfected fish. The moribund fish exhibited (**b**) loss of color and fin hemorrhage (red arrowheads); (**c**) hemorrhage on the body; (**d**) lesions around the mouth; (**e**) reddening of the anus; (**f**) exophthalmia; and (**g**) hemorrhage in the eye. The scale bar was added using ImageJ software; (**h**) daily survival rate of *V. anguillarum* J382 infected lumpfish. Colonization of *V. anguillarum* J382 in lumpfish organs at 5 dpi using (**i**) 10^6^ CFU/mL and (**j**) 10^7^ CFU/mL.

**Figure 2 microorganisms-11-00792-f002:**
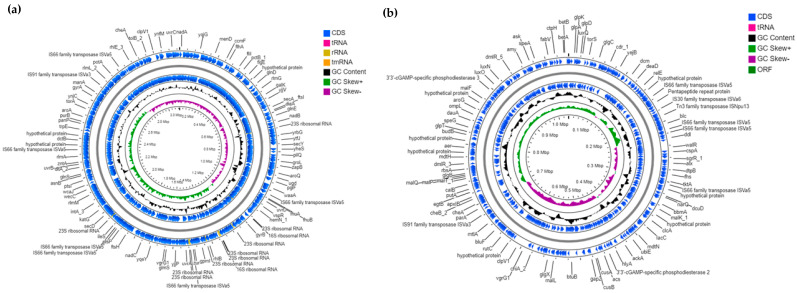
Genomic maps and some features of the *V. anguillarum* J382 chromosomes. (**a**) Genomic map of *V. anguillarum* J382 chromosome 1, and (**b**) genomic map of *V. anguillarum* J382 chromosome 2. The maps were drawn using CGView BETA Server and the annotation was carried out using Prokka software implemented in CGView.

**Figure 3 microorganisms-11-00792-f003:**
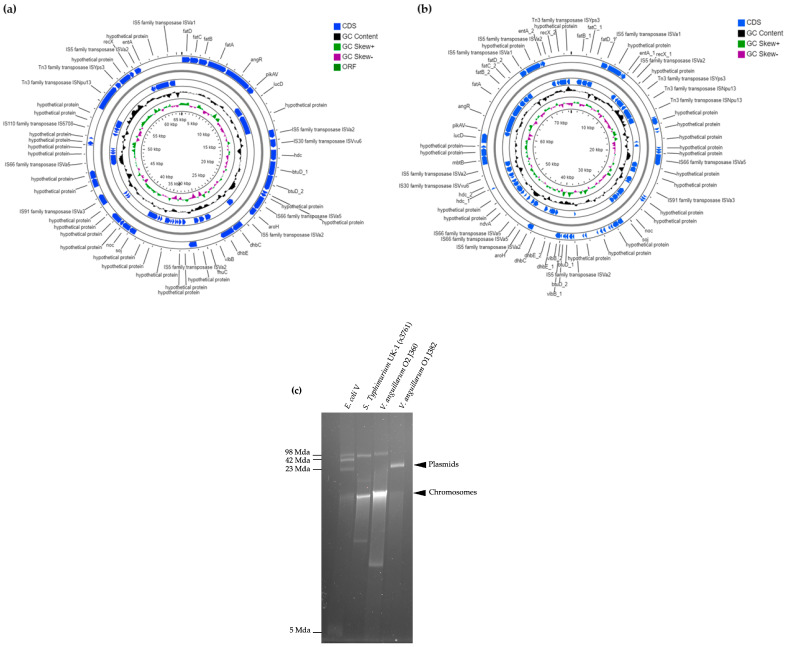
Genomic maps and some features of the *V. anguillarum* J382 plasmids. (**a**) Genomic map of plasmid pVA_O1_1. (**b**) Genomic map of plasmid pVA_O1_2. The maps were drawn using CGView BETA Server and the annotation was carried out using Prokka software implemented in CGView. (**c**) Plasmid extraction using the plasmid alkaline extraction method. For markers, the bacteria *Escherichia coli* (*E. coli*) V and *Salmonella Typhimurium* (*S. Typhimurium*) UK-1 (χ3761) were used, while *V. anguillarum* J360 was used for comparison.

**Figure 4 microorganisms-11-00792-f004:**
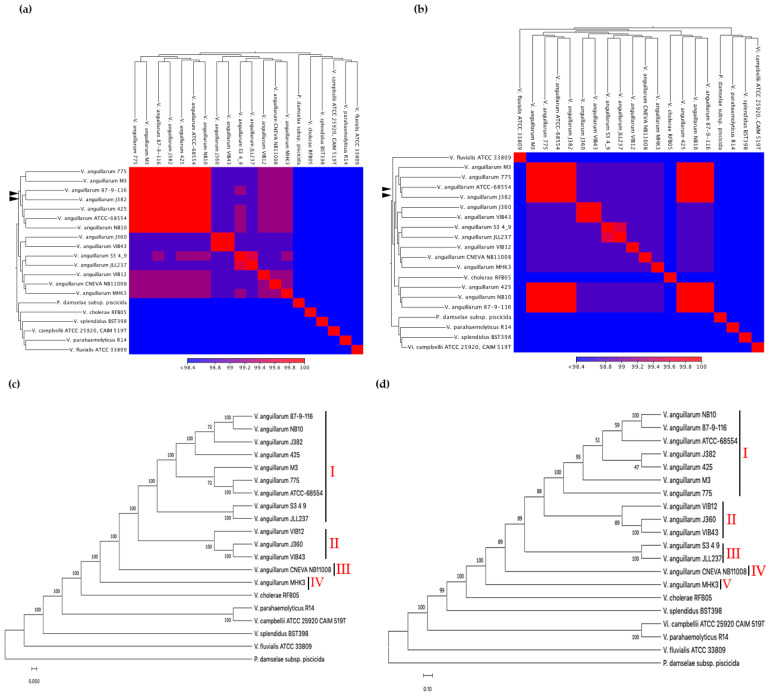
Average nucleotide identities (ANI) were computed for (**a**) *V. anguillarum* J382 chromosome 1, and (**b**) *V. anguillarum* J382 chromosome 2 in comparison with complete genomes of *V. anguillarum* serotypes O1, O2, O3, and related *Vibrio* species. The *V. anguillarum* J382 chromosomes and their closest counterparts are indicated by black arrow heads. The whole genome phylogenetic tree for (**c**) *V. anguillarum* J382 chromosome 1, and (**d**) *V. anguillarum* J382 chromosome 2. The phylogeny was inferred using the neighbor-joining method, the Kimura method was used for computing the evolutionary distance, and 1000 bootstrap replicates were used. *P. damselae* subsp. piscicida was used as the outgroup.

**Figure 5 microorganisms-11-00792-f005:**
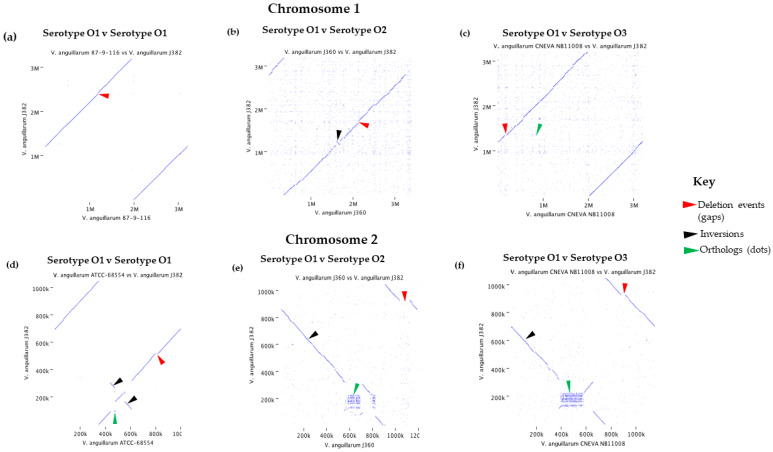
Synteny plots for *V. anguillarum* J382 chromosome 1 against (**a**) serovar O1 *V. anguillarum* 87-9-116; (**b**) serovar O2 *V. anguillarum* J360; (**c**) serovar O3 *V. anguillarum* CNEVA NB 11008. Synteny plots for *V. anguillarum* J382 chromosome 2 against; (**d**) serovar O1 *V. anguillarum* 87-9-116; (**e**) serovar O2 *V. anguillarum* J360; (**f**) serovar O3 *V. anguillarum* CNEVA NB 11008.

**Figure 6 microorganisms-11-00792-f006:**
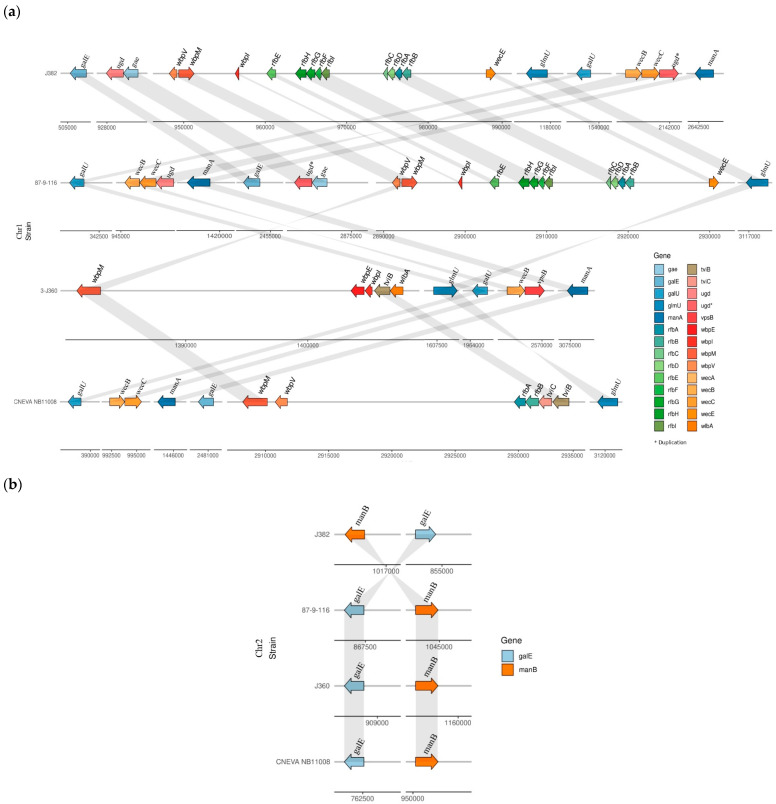
O-antigen biosynthesis gene cluster diversity among *V. anguillarum* serotypes O1, O2, and O3. (**a**) O-antigen gene arrangement in chromosome 1; *: gene duplication (**b**) O-antigen gene arrangement in chromosome 2.

**Figure 7 microorganisms-11-00792-f007:**
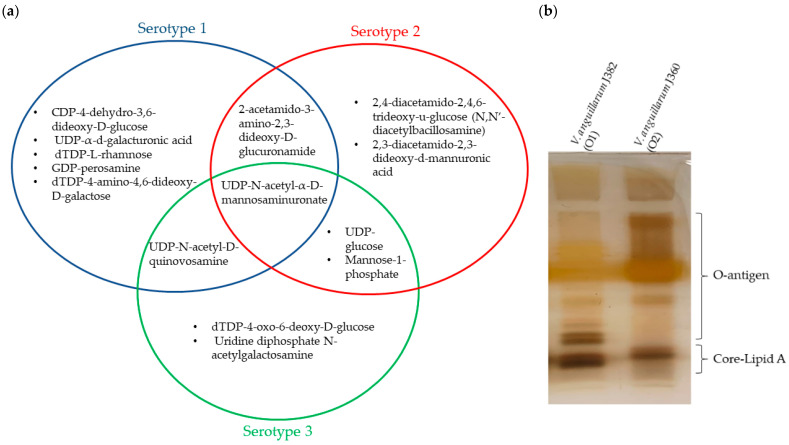
LPS characterization of *V. anguillarum* serotypes O1, O2, and O3. (**a**) Signature oligosaccharides that correspond to the sugar synthesis pathways identified using the KEGG platform in *V. anguillarum* serotypes O1, O2, and O3. (**b**) LPS profiles of *V. anguillarum* J382 (serotype O1) and *V. anguillarum* J360 (serotype O2). SDS-PAGE 15%; LPS bands were separated at 120V for 120 min.

**Figure 8 microorganisms-11-00792-f008:**
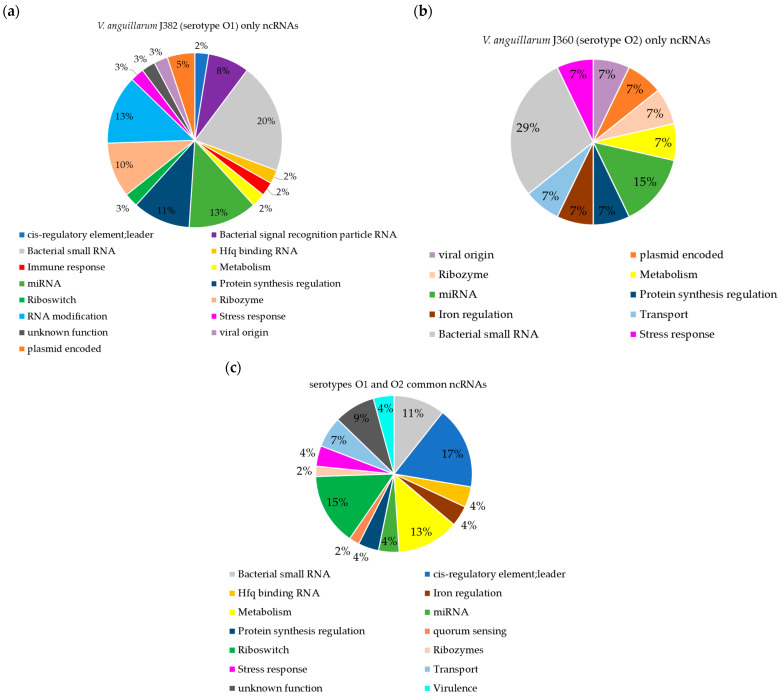
Comparison of ncRNAs in *V. anguillarum* serotypes O1 (J382) and O2 (J360). (**a**) ncRNAs present only in *V. anguillarum* J382 (serotype O1). (**b**) ncRNAs present only in *V. anguillarum* J360 (serotype O2). (**c**) commonly shared ncRNAs in *V. anguillarum* J382 and *V. anguillarum* J360. The ncRNAs for *V. anguillarum* J382 were predicted using the software StructRNAfinder and BEDTools: intersect v2.3, while J360 ncRNAs were retrieved from the NCBI. The ncRNA comparison was performed using jvenn software.

**Table 1 microorganisms-11-00792-t001:** Genomes used in this study.

Strain (Serotype)	Host (Year)	Geographical Location	Accession	Size (bp)	GC%
*V. anguillarum* J382 (O1)	*Oncorhynchus mykiss irideus* (1999)	Canada: British Columbia	CP091185/CP091186	4,171,832	44.34
*V. anguillarum* 775 (O1)	*Oncorhynchus kisutch* (2011)	USA: Pacific Ocean coast	CP002284.1/CP002285.1	4,052,047	44.48
*V. anguillarum* ATCC-68554 (O1)	*Oncorhynchus kisutch* (1977)	USA: Pacific Ocean coast	CP023208.1/CP023209.1	4,141,910	44.52
*V. anguillarum* VIB43 (O1)	*Dicentrarchus labrax*	UK: Scotland	CP023054.1/CP023055.1	4,407,865	44.54
*V. anguillarum* JLL237 (O1)	*Oncorhynchus mykiss* (1995)	Denmark	CP022101.1/CP022102.1	4,286,989	44.48
*V. anguillarum*87-9-116 (O1)	*Salmo salar* (1987)	Finland	CP021980.1/CP021981.1	4,338,125	44.34
*V. anguillarum* M3 (O1)	*Paralichthys olivaceus* (1999)	China: Shandong	CP006699.1/CP006700.1	4,117,885	44.45
*V. anguillarum* NB10 (O1)	*Oncorhynchus mykiss* (2016)	Sweden: Baltic Sea, Norrbyn Umeaa	LK021130.1/LK021129.1	4,373,835	44.37
*V. anguillarum* J360 (O2)	*Cyclopterus lumpus* (2018)	Canada: St John’s, Newfoundland	CP034672.1/CP034673.1	4,549,570	44.46
*V. anguillarum* VIB12 (O2)	*Dicentrarchus labrax*	Greece	CP023310.1/CP023311.1	4,897,690	44.46
*V. anguillarum* MHK3 (other)	*Paralichthys olivaceus* (2006)	China: Weihai	CP022468.1/CP022469.1	4,015,925	44.71
*V. anguillarum*425 (O1)	*Dicentrarchus labrax* (1999)	China: Yellow Sea	CP020534.1/CP020533.1	4,373,373	44.41
*V. anguillarum* S3 4/9 (O1)	*Oncorhynchus mykiss* (1995)	Denmark	CP022099.1/CP022100.1	4,182,973	44.56
*V. anguillarum* CNEVA NB11008 (O3)	*Dicentrarchus labrax* (1997)	France: Brest	CP022103.1/CP022104.1	4,256,429	44.56
*V. fluvialis* ATCC 33809	*Homo sapiens*	Bangladesh: Dacca	CP014034.2/CP014035.2	4,827,733	49.90
*V. parahaemolyticus* R14	*Penaeus vannamei* (2016)	Pacific Ocean	CP028141.1/CP028142.1	5,444,136	45.27
*V. cholerae* RFB05 (O1)	Freshwater (2017)	Water USA: Pittsburgh	CP043557.1/CP043558.1	4,357,322	47.03
*V. splendidus* BST398	coastal sea water (2015)	North Pacific Ocean	CP031055.1/CP031056.1	5,508,387	44.12
*V. campbellii* ATCC 25920, CAIM 519T	Ocean water (1972)	USA: Hawaii	CP015863.1/CP015864.1	5,178,103	45.09
*Photobacterium damselae* subsp. Piscicida91-197	*Morone chrysops* × *Morone saxatilis*	USA	AP018045.1/AP018046.1	4,293,175	41.01

**Table 2 microorganisms-11-00792-t002:** Phenotypic characteristics of *V. anguillarum* J382 isolated from *Oncorhynchus mykiss* using phenotypic tests and commercial biochemical test kits.

Phenotypic Test		API20NE		API20E		APIZYM	
Hemolysis	+	Reduction of nitrates to nitrites	+	β-Galactosidase	+	Alkaline phosphatase	+
Motility	+	Indole production	+	Indole production	+	Esterase (C_4_)	+
Siderophore synthesis	+	Glucose fermentation	+	Acetoin production	+	Esterase lipase (C_8_)	+
O/F glucose	+	Urease	-	Urease	-	Lipase (C_14_)	+
O/F arabinose	+			H_2_S production	-	Leucine arylamidase	+
Catalase	+			Citrate utilization	+	Valine arylamidase	+
Oxidase	+	**Hydrolysis of:**		**Hydrolysis of:**		Cystine arylamidase	+
**Growth at:**		Arginine	+	L-Arginine	+	Trypsin	-
4 °C	+	Esculin	+	L-Lysine	-	α-Chymotrypsin	-
15 °C	+++	Gelatin	+	Gelatin	+	Acid phosphatase	+
28 °C	+++			L-Tryptophane	-	Naphthol-AS-BI Phosphohydrolase	+
37 °C	-			L-Ornithine	-	α-Galactosidase	-
LB NaCl 0%	-	**Assimilation of:**		**Assimilation of**		β-Galactosidase	+
LB NaCl 0.5%	+	D-Glucose	+	D-Glucose	+	β-Glucoronidase	-
LB NaCl 2%	+	L-Arabinose	+	L-Arabinose	+	α-Glucosidase	+
Plate Count Agar 50% seawater	+	D-Mannose	+	Inositol	-	β-Glucosidase	-
**Antibiogram:**	Halo diameter (mm)	D-Mannitol	+	D-Mannitol	+	N-Acetyl-β-glucosaminidase	+
Vibriostatic agent (0/129)	22 (sensitive)	N-acetyl-glucosamine	+	L-Rhamnose	-	α-Mannosidase	-
Tetracycline (10 µg)	39 (sensitive)	D-maltose	-	D-Saccharose	+	α-Fucosidase	-
Oxytetracycline (30 µg)	40 (sensitive)	Potassium gluconate	+	D-Melibiose	-		
Ampicillin (10 µg)	0 (resistant)	Capric acid	-	D-Amygdaline	+		
Sulfamethoxazole (25 µg)	25 (sensitive)	Adipic acid	-	D-Sorbitol	+		
Chloramphenicol (30 µg)	32 (sensitive)	Malic acid	+				
Colistin sulphate (10 µg)	0 (resistant)	Trisodium citrate	-				
Oxalinic acid (2 µg)	38 (sensitive)	Phenylacetic acid	-				

+: growth or positive reaction; +++: substantial growth; -: negative growth or reaction.

## Data Availability

Not applicable.

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
