# Peer review of "Comparative Genomic Analysis of Virulent Vibrio (Listonella) anguillarum Serotypes Revealed Genetic Diversity and Genomic Signatures in the O-Antigen Biosynthesis Gene Cluster"

_microorganisms, 2023, doi:10.3390/microorganisms11030792_

Round 1

Reviewer 1 Report

Comments to authors:

-The current study is interesting; however, the authors should address the following comments to improve the quality of the manuscript:

-The manuscript should be revised for English editing and grammar mistakes.

- Write the scientific names of bacterial pathogens and genes in the correct form all over the manuscript and the references section.

Title:

I think the work would benefit from the title that contains the main conclusion of the study (should be derived from the conclusion). Please modify the title.

For example, the authors compared their retrieved isolate with serotypes O1, O2 and O3, however in abstract there is no real comparative with O3 serotype. Please clarify

Abstract:

-  The abstract must illustrate the implemented methods such as software analysis tools and the most predominant phylogenetic results. Besides, rephrase the aim of the work and the main conclusion of your findings.

-A graphical abstract is recommended (If possible).

- Lines 24 and 27 reflect ambiguity and misunderstanding, as the author firstly mentioned that V. anguillarum O1 isolated from a winter Steelhead trout (Oncorhynchus mykiss irideus) then later said it caused 100% mortality in naïve lumpfish (Cyclopterus lumpus), please clarify?

Introduction: (it needs to be more informative):

- Reference 1 is a recent report 2011, so how it explain that the following statement (Vibrio anguillarum (Listonella anguillarum) have been recorded as early as 1893. Please revise and modify

- Lines 44-46 showed some sort of redundancy with the lines 49-51, please review   

-Give a hint about the virulence factors and the mechanism of disease occurrence, stress mediated factors, and antibiotic resistance of that pathogen you can follows:

https://doi.org/10.1016/j.aquaculture.2021.736447

DOI 10.3389/fmicb.2023.1135614

- Line 83: please revise until recently to until now

- Lines 90-94 seem more related to the result section, please concise to magnify only the purpose of study and aim of work. Rephrase the aim of the work to be clear and better sound.

Materials and methods:

- Support all methods with updated specific references.

- start line 97 with Vibrio anguillarum

- Add the company, city, and country of the used chemicals and reagents.

- Lines 108–113 are repeated downwards and explain more physiological characteristics of bacteria than the cultivation condition, please delete.

-line 141: API ZYM strips were known to be used for flavobacterium and API20NE for members of the enterobacteriaceae, so why the author used them

- Line 155: How the authors know that the selected fish are Specific pathogen free. Did they perform any microbiological examinations, or used genetically modified species with a certificate.

- 55% CP is more than the optimal requirement for fish feed and may increase the level of ammonia produced, so please clarify the following:

Did you monitor the ammonia level meanwhile the adaptation and what is the suitable protocol for that?

What about the feeding frequency per day?

- Line 168: fish were acclimatized again for 2 weeks under optimal conditions. Is it correct?

- Line 172: why the author used bathing route of infection, not IM or IP, support your challenge with a reference

- Line 176: as non-infected control and were immersed with 1x PBS.

Be aware that immersion is completely different from bathing (different exposure times), the three tanks should exposed to the same condition

- Move table 1 to the section Genomes used in this study

- Line 195: revise as genomic DNA

- What kind of primers used for sequencing

- please add 2.12 section of statistical analysis as a separate section

- Line 310: p should be italic

- 312: use only tissue not tissue organ

•The authors are advised to classify the tested isolates to MDR , XDR, and PDR as described by Magiorakos et al.

Magiorakos AP, Srinivasan A, Carey RB, Carmeli Y, Falagas ME, Giske CG, et al. Multidrug-resistant, extensively drug-resistant and pandrug-resistant bacteria: An international expert proposal for interim standard definitions for acquired resistance. Clin Microbiol Infect. 2012; 18:268–81. doi:10.1111/j.1469-0691.2011.03570.x.

Discussion:

- The authors are advised to illustrate the real impact of their findings without repetition of results.

- Please illustrate different mechanisms of antibacterial resistance in these bacteria

Conclusion

- Should be concise. A real conclusion should focus on the question or claim you articulated in your study, which resolution has been the main objective of your paper?

Author Response

Reviewer #1

Comments and Suggestions for Authors

Comments to authors:

-The current study is interesting; however, the authors should address the following comments to improve the quality of the manuscript:

-The manuscript should be revised for English editing and grammar mistakes.

- Write the scientific names of bacterial pathogens and genes in the correct form all over the manuscript and the references section.

Title:

I think the work would benefit from the title that contains the main conclusion of the study (should be derived from the conclusion). Please modify the title.

For example, the authors compared their retrieved isolate with serotypes O1, O2 and O3, however in abstract there is no real comparative with O3 serotype. Please clarify.

RE: We modified the title to “Comparative genomic analysis of Vibrio anguillarum revealed genetic diversity and genomic signatures in the O-antigen biosynthesis gene cluster of virulent serotypes”

Abstract:

-  The abstract must illustrate the implemented methods such as software analysis tools and the most predominant phylogenetic results. Besides, rephrase the aim of the work and the main conclusion of your findings.

-A graphical abstract is recommended (If possible).

- Lines 24 and 27 reflect ambiguity and misunderstanding, as the author firstly mentioned that V. anguillarum O1 isolated from a winter Steelhead trout (Oncorhynchus mykiss irideus) then later said it caused 100% mortality in naïve lumpfish (Cyclopterus lumpus), please clarify?

RE: for better understanding the abstract was modified to:

Vibrio anguillarum serotypes O1, O2 and O3 are the only virulent serotypes and the most frequent pathogen affecting fish world-wide. Genetic differences between serotypes that could shed insight on the evolution and serotype differences of this marine pathogen are unknown. Here, we fully sequenced and characterized a strain of V. anguillarum O1 (J382) isolated from a winter Steelhead trout (Oncorhynchus mykiss irideus) in British Colombia, Canada. Koch postulates using V. anguillarum J382 strain were replicated in naïve lumpfish (Cyclopterus lumpus), which caused lethal vibriosis within 7 dpi. Phenotypic and genotypic comparisons were done for the three virulent serotypes in fish using biochemical tests and bioinformatic tools, respectively. The genome of V. anguillarum J382 contains two chromosomes (3,13 Mb and 1,03 Mb) and two typical pJM1-like plasmids (65,573 and 76,959 bp). Furthermore, V. anguillarum O1 displayed resistance to colistin sulphate which differs from serotype O2 and could be attributed to the presence of the ugd gene. Comparative genomic analysis among the serotypes showed that intra-species evolution is driven by insertion sequences, bacteriophages, and a different repertoire of putative ncRNAs. Genetic heterogeneity in the O-antigen biosynthesis gene cluster is characterized by the absence or the presence of unique genes, which could result in differences in immune evasion mechanisms employed by the respective serotypes. This study contributes towards understanding the genetic differences among V. anguillarum serovars and their evolution.

Introduction: (it needs to be more informative):

- Reference 1 is a recent report 2011, so how it explain that the following statement (Vibrio anguillarum (Listonella anguillarum) have been recorded as early as 1893. Please revise and modify

RE: The original reference “Canestrini G. La malattia dominate delle anguille, 1893. Atti Ist Veneto Sci Lett Arti Cl Sci Mat Nat 7, 809–814.” And the text was modified to: “Reports of vibriosis in freshwater, brackish water and marine aquatic animals caused by the Gram-negative bacterium, commonly known as Vibrio anguillarum (Listonella anguillarum) have been recorded as early as 1893 and continue today [1, 2].”

- Lines 44-46 showed some sort of redundancy with the lines 49-51, please review   

RE: The sentence ‘These fish mortalities result in substantial economic losses to the industry.’, was deleted.

-Give a hint about the virulence factors and the mechanism of disease occurrence, stress mediated factors, and antibiotic resistance of that pathogen you can follows:

https://doi.org/10.1016/j.aquaculture.2021.736447

DOI 10.3389/fmicb.2023.1135614

RE: The present article is focus on comparative genomic, but we have cited excellent reviews on V. anguillarum physiology, virulence factors and epidemiology. Please see references 2-5, 54.

The suggested reference “Inevitable impact of some environmental stressors on the frequency and pathogenicity of marine vibriosis” is very generic and far from the aim of our article, thus is not supporting our study and was not cited. The suggested article “Editorial: Emerging multidrug-resistant bacterial pathogens "superbugs": A rising public health threat” is also far from the objective of this article and specific article on V. anguillarium antibiotic resistance are cited in the article. Please see reference 26, 55.

- Line 83: please revise until recently to until now

RE: the sentence was revised accordingly and the text was modified to: “V. anguillarum is composed of 23 O-serotypes but typically only serotypes O1, O2 and O3 are associated with vibriosis in ]aquatic animals while other serotypes are non-pathogenic environmental isolates [6]. Biochemically, V. anguillarum is halophilic, catalase and oxidase positive, lysine and ornithine decarboxylase negative, ferments D-glucose without gas production, reduces nitrate to nitrite, is sensitive to the vibriostatic agent 0/129 and cannot ferment inositol or rhamnose [7]. Phenotypic characterization includes motility, growth below 30°C, non-spore-forming, and facultatively anaerobic metabolism [1, 6, 8]. Additionally, extracellular enzyme activities for caseinase, lipase, phospholipase, hemolysin, gelatinase and siderophore synthesis were recorded for most serotype O1, O2 and O3 with a few exceptions [9]. On the genomic level, this bacterium contains two chromosomes which are around 3.0 and 1.2 Mbp in size and GC content of 43–46% [10]. Some serotypes O1 and O2 isolates have also been reported to contain the virulence plasmids that are 65–67 kbp and are similar to plasmid pJM1 that contains the genes encoding a siderophore-dependent iron sequestering system [11, 12] and at times to contain a smaller plasmid around 12 kbp [5].

 In the Pacific Northwest, the steelhead trout (Oncorhynchus mykiss irideus) is an important fish species for the local economy, culture, and ecosystems [17]. Steelhead trout are affected by several infectious agents like hematopoietic necrosis virus (IHNV) [18] and co-infections of IHNV and Flavobacterium psychrophilum [19]. Other diseases reported in steelhead trout include parasitic black spot disease [20] and enteric red mouth disease (ERM) caused by Yersinia ruckeri [21]. Currently, there have being no reports of V. anguillarum infection in steelhead trout, although infections have been reported in closely related sea-reared rainbow trout [22].

- Lines 90-94 seem more related to the result section, please concise to magnify only the purpose of study and aim of work. Rephrase the aim of the work to be clear and better sound.

RE: the text was modified to:

In 1999, a strain of V. anguillarum O1, designated here J382, was isolated a winter Steelhead trout on the Little Campbell River, British Colombia, Canada [23] and used in the present study. This study therefore aimed to characterize this strain and to describe the characteristics of the complete genome. Phenotypic, biochemical, and genomic characterization confirmed that V. anguillarum J382 strain is serotype O1, closely related to other V. anguillarum O1 strains. Virulence was determined in lumpfish. Comparative genomics of V. anguillarum serotypes O1, O2 and O3 revealed genetic diversity in the O-antigen biosynthesis gene cluster

Materials and methods:

- Support all methods with updated specific references.

RE: All methods are supported with updated specific references. Please see the updated manuscript.

- start line 97 with Vibrio anguillarum

RE: All methods are supported with updated specific references. Please see the updated manuscript.

- Add the company, city, and country of the used chemicals and reagents.

RE: Where possible, the company, city and country of the chemicals and reagents used were added. Please see the updated manuscript.

- Lines 108–113 are repeated downwards and explain more physiological characteristics of bacteria than the cultivation condition, please delete.

RE: The lines are relevant to section as they describe different culture conditions for the following experiments. The cultivation conditions are explained in full in section 2.2.

-line 141: API ZYM strips were known to be used for flavobacterium and API20NE for members of the enterobacteriaceae, so why the author used them

RE: They are also known to be used for several bacterial species, including Vibrio anguillaruim. API ZYM, NE are for Gram negative bacteria and V. anguillarum is Gram negative, thus we used to compare between strains and the published literature.

- Line 155: How the authors know that the selected fish are Specific pathogen free. Did they perform any microbiological examinations, or used genetically modified species with a certificate.

RE: The fish facility and the fish are inspected each quarter by the provincial authorities and tested for reportable diseases and non-reportable diseases, including V. anguillarum. Prescence of these pathogens will cause the elimination of all the cohort. In addition, the seawater is intake at 60 mt deep, sand-filterd, UV treated, and degassed. The fish are produced from healthy cultivated broodstocks by in vitro fertilization under aseptic conditions. These measurements allow to have a facility specific pathogens free lumpfish.

- 55% CP is more than the optimal requirement for fish feed and may increase the level of ammonia produced, so please clarify the following:

Did you monitor the ammonia level meanwhile the adaptation and what is the suitable protocol for that?

RE: As we mentioned in the manuscript, our facility is flowthrough, ammonia and other toxins are no accumulated and optimal conditions are maintained. We do not use static tanks that are not suitable for state-of-the-art fish experimentation.

What about the feeding frequency per day?

RE: 3 times per day 0.5% body weight at indicated in the text.

- Line 168: fish were acclimatized again for 2 weeks under optimal conditions. Is it correct?

RE: Yes. The optimal conditions for lumpfish culture are stated in section 2.3 and 2.4.

- Line 172: why the author used bathing route of infection, not IM or IP, support your challenge with a reference

RE: Bath route of infection was used because it mimics natural infection with short exposure times. We have optimized methods to more natural infection process.

- Line 176: as non-infected control and were immersed with 1x PBS. Be aware that immersion is completely different from bathing (different exposure times), the three tanks should exposed to the same condition

RE: The text was modified to: “Quantification of the bacterial was carried out by plate count method. Lumpfish in tanks 1 and 2 were bath infected for 30 mins with freshly prepared 106 CFU/mL and 107 CFU/mL bacterial inoculum of V. anguillarum J382, respectively, while lumpfish in tank 3 were used as non-infected control and were immersed in seawater.”

- Move table 1 to the section Genomes used in this study

RE: The template of the journal makes its difficult to put the table directly underneath the section as the table is big. Hence, the table was place as close to the section as possible.

Line 195: revise as genomic DNA

RE: The sentence was revised, please see update version

- What kind of primers used for sequencing

RE: We do know what the reviewer is referring to. We used next generation sequencing short reads and long reads. Illumina, or short read sequencing use adaptors, perhaps is that what the reviewer referrers, but these adaptors are included in the illumina kit for library preparations and is was cited int eh text. The PacBio technology does not use primers.

- please add 2.12 section of statistical analysis as a separate section

RE: Statistical analysis was presented as a separate section. See section 2.12

- Line 310: p should be italic

RE: corrected, please see up dated text version.

- 312: use only tissue not tissue organ

RE: corrected

  • The authors are advised to classify the tested isolates to MDR , XDR, and PDR as described by Magiorakos et al.

Magiorakos AP, Srinivasan A, Carey RB, Carmeli Y, Falagas ME, Giske CG, et al. Multidrug-resistant, extensively drug-resistant and pandrug-resistant bacteria: An international expert proposal for interim standard definitions for acquired resistance. Clin Microbiol Infect. 2012; 18:268–81. doi:10.1111/j.1469-0691.2011.03570.x.

RE: This article is about V. anguillarum genomics not about antibiotic resistance. We cannot really apply this classification and redirect the manuscript towards a different field that the reviewer is suggesting.

Discussion:

The authors are advised to illustrate the real impact of their findings without repetition of results.

Please illustrate different mechanisms of antibacterial resistance in these bacteria

RE: The discussion was modified as suggested. Please see the up dated version

Conclusion

- Should be concise. A real conclusion should focus on the question or claim you articulated in your study, which resolution has been the main objective of your paper?

RE: The conclusion was modified as suggested, focusing on comparative genomics:

“V. anguillarum J382 was isolated from a natural infection in winter Steelhead trout in the Pacific coast and subjected to phenotypic and genomic characterization. In a laboratory challenge, the bacterial isolate displayed pathogenicity in Atlantic lumpfish. Phenotypic characterization also revealed that V. anguillarum J382 is resistant to colistin sulphate whilst a serotype O2 (V. anguillarum J360) is susceptible. This colistin resistance is probably conferred by the ugd gene present in serotype O1 and absent in serotype O2 which after being phosphorylated was implicated in resistance to cationic antimicrobial peptides and antibiotics such as polymyxin and colistin. Comparative genomics in the O-antigen gene clusters of serotypes O1, O2 and O3 revealed genetic variation in gene composition and arrangement, O-antigen translocation and polymerization and oligosaccharide composition. ANI, phylogeny and synteny revealed the closest relatives of V. anguillarum J382 as well as genetic diversity among the different serotypes showing intra-species evolution. Some of the O-antigen biosynthesis pathways identified were earlier identified in distantly related Gram-negative bacteria such as P. aeruginosa indicating inter-species lateral transfer of the O-antigen gene clusters. ncRNAs were also annotated and they provided insight into the various putative functions these might carry out in V. anguillarum under physiological adaptation. Future work will focus on characterization of the O-antigen composition of the three serotypes using spectroscopy techniques that can elucidate the physical, chemical, electronic and structural information. Also, it would be interesting to study the effects of gene knock-down of some of the O-antigen genes and the potential application of the resulting mutants as attenuated vaccine candidates. Similarly, potential of O-antigen components as subunit vaccines needs to be studied since the O-antigen have immunomodulatory effects in the host. Our work contributes towards understanding pathogenesis of V. anguillarum as well as laying the foundation vaccine development via manipulation of the O-antigen gene clusters.”

Reviewer 2 Report

The manuscript is well organized designed and written here you can find only minor comments

1.       Line 195 "" intergrity" spelling checking

2.       The following link is not leading to a server please check  http://stothard.afns.ualberta.ca/cgview_server/    

3.       Line 227 Do you mean Proksee or PROKKA

4.       Line 275 " ealier" spell check

5.       The figures resolutions are poor try to increase it

6.       Check the discussion some gen names are not italic

Author Response

Reviewer #2

Comments and Suggestions for Authors

The manuscript is well organized designed and written here you can find only minor comments

  1. Line 195 "" intergrity" spelling checking
  2. The following link is not leading to a server please check  http://stothard.afns.ualberta.ca/cgview_server/    
  3. Line 227 Do you mean Proksee or PROKKA
  4. Line 275 " ealier" spell check
  5. The figures resolutions are poor try to increase it
  6. Check the discussion some gen names are not italic

RE: All the suggestion/edited were considered. Thanks!

Reviewer 3 Report

1-    Keywords should be changed to include words that described the research and not present in the title as: Whole genome sequencing; winter Steelhead trout; Pathogenicity; biochemical identification, Vibriosis, lumpfish

2-    Chloramphenicol is paned for use worldwide and Oxalinic acid not currently used

Author Response

Reviewer #3

Comments and Suggestions for Authors

  • Keywords should be changed to include words that described the research and not present in the title as: Whole genome sequencing; winter Steelhead trout; Pathogenicity; biochemical identification, Vibriosis, lumpfish

RE: The key words were modified to: “Whole genome sequencing; Vibrio anguillarum; winter Steelhead trout; biochemical identification, vibriosis, lumpfish; pathogenicity”

2-    Chloramphenicol is paned for use worldwide and Oxalinic acid not currently used

      RE: Although the two antibiotics were either banned of discontinued, they were used in this study because they were used in previous study that focused on antibiotic resistance of V. anguillarum serovars. This helped in the characterization of V. anguillarum J382.Therefore, the following sentence was added to section 2.2.4 to clarify why the antibiotics were used:

Choice of antibiotics tested was based on previous antibiotic resistance study in V. anguillarum serovars [55].

Reviewer 4 Report

My comments are here:

Line 103, “Routine cultures” – explain it in detail.

Line 104, how many hours took to reach mid-logarithmic phase.

Line 108, it is not clarified, why 2,2-dipyridyl and FeCl3 were added in the medium.

Line 112, only hemolytic activity was demonstrated but not salmon blood, Any explanation.

Line 155, how fish were confirmed as Specific pathogen free and how many fish were used for this purpose.

Figure 1: Please add scale bar for the fish.

Please improved the quality of figures and increase the font size in Figures 2, 3, 4, 5, 7 and 8.

Author Response

Reviewer #4

Comments and Suggestions for Authors

My comments are here:

Line 103, “Routine cultures” – explain it in detail.

RE: Routine cultures are grown using standard protocols for bacterial isolation and culture. The standard protocol used for reviving the cryopreserved bacteria and routine culture were the same. Hence the sentences were modified as follows:

The bacterium was revived from cryopreservation and routinely grown on Trypticase Soy Agar (TSA; Difco, Franklin Lakes, NJ, USA) supplemented with 2% NaCl and 1.5 % bacto agar (Difco) and incubated for 48 h at 15°C [5]. Routine cultures produced following standard procedures as described above were maintained in 3 mL of Trypticase Soy Broth (TSB; Difco, Franklin Lakes, NJ, USA) supplemented with 2% NaCl at 15°C for 18-24 h until mid-logarithmic phase (optical density (OD600) ≈ 0.7 (~ 4.1 x 108 CFU/mL)).

Line 104, how many hours took to reach mid-logarithmic phase.

RE: It took 18-24 h to reach mid-logarithmic phase. Please see the response above.

Line 108, it is not clarified, why 2,2-dipyridyl and FeCl3 were added in the medium.

Response:  The sentence was modified to explain the addition of 2,2-dipyridyl and FeCl3. ‘When required, TSB was supplemented with 100 µM 2,2-dipyridyl to deplete iron in the media and 100 µM of FeCl3 for iron enrichment in the media.’

Line 112, only hemolytic activity was demonstrated but not salmon blood, Any explanation.

RE: The method in section 2.1 was edited and the phrase ‘salmon blood’ was deleted since we were unable to obtain fresh salmon blood for hemolytic activity. The sentence now reads:

‘Bacterial hemolytic activity was evaluated in TSA plates supplemented with 5% of sheep blood.’

Line 155, how fish were confirmed as Specific pathogen free and how many fish were used for this purpose.

RE: The fish facility and the fish are inspected each quarter by the provincial authorities and tested for reportable diseases and non-reportable diseases, including V. anguillarum. Prescence of these pathogens will cause the elimination of all the cohort. In addition, the seawater is intake at 60 mt deep, sand-filterd, UV treated, and degassed. The fish are produced from healthy cultivated broodstocks by in vitro fertilization under aseptic conditions. These measurements allow to have a facility specific pathogens free lumpfish.

Figure 1: Please add scale bar for the fish.

RE: Scale bar was added. Please see Figure 1. The following sentence was added in the caption, ‘Scale bar was added using imagej software.’

Please improved the quality of figures and increase the font size in Figures 2, 3, 4, 5, 7 and 8.

RE: Higher quality imaging will be provided in the final version as the weight is in the upper sized for cut off. We will follow all the journal instruction for figure quality.

Round 2

Reviewer 1 Report

Based on initial evaluation of the revised version: The authors have revised the entire manuscript and addresses all scientific inquiries and comments. However, they should put in more effort to correct the many grammatical and typographical errors and to keep the conclusion part brief. 

Line 21: please revise as the most, Worldwide, Known

Add comma before and in the following lines 22, 27, 51, 59, 91, 170, 767

Line 23:  revise as Insight into

Line 25: remove letter (a) before winter

Line 54: revise as oxidase-positive

Line 82: revise as have been

Line 85: revise as isolated from

Line 87: revise as  describe

Line 165: the authors are encouraged to add the following information provided in the rebuttal letter to prove that the fish is Specific pathogen free.

- regarding the feeding frequency: 3 times per day statement was not included, please add

- Line 168: n= should be in italic form

- Line 171: revise as described and the mid logarithmic phase

- Line 180: remove comma after ms222 and add comma before and spleen

- Line 181: revise as the weight

- Line 720: unneeded space 

- Line 738: revise as was done

- Line 744: revise as with an N-formyl-L-alanyl group

Line 750: revise as the absence

Line 760: revise as the discovery

Authors are advised to concise the conclusion part, as it contains many sentences related to the result section. They should present only the important results and maximize their importance for the future

Author Response

Editors Comment:

I am writing this email is to ask for your kind provision of supplementary Figure S3d-f.

As we noticed, it is mentioned in the main text, Section 3.5.2. Synteny analysis, however,

the subfigures are not found in your supplementary file. We are highly appreciated if you

could provide us the supplementary subfigures as soon as possible.

RE: The indication of Figure S3d-f in Section 3.5.2. Synteny analysis was a typing error. This have been corrected to Figure S4d-f in Lines 462-472. The section now reads as follows:

Comparisons between the chromosomes of V. anguillarum J382 with those of con-specific isolates were performed to study synteny. Dot plot visualization of serotype O1 chromosome 1 (V. anguillarum J382 vs V. anguillarum 87-9-116) showed high similarity (Figure 5a), although there is genomic rearrangement of the locally collinear blocks (Figure S4a). Visualization of V. anguillarum J382 chromosome 1 against V. anguillarum J360 chromosome 1 (serotype O2) (Figure 5b) and V. anguillarum J382 chromosome 1 against V. anguillarum CNEVA NB 11008 chromosome 1 (serotype O3) (Figure 5c) revealed genomic gaps, inversions and orthologs. This was supported by the locally collinear blocks arrangements that also showed genomic rearrangements (Figure S4b and S4c). Similar results were observed when synteny was studied for V. anguillarum J382 chromosome 2 (Figures 5d-f and Figure S4d-f).

Reviewer 1 Comments and responses

Based on initial evaluation of the revised version: The authors have revised the entire manuscript and addresses all scientific inquiries and comments. However, they should put in more effort to correct the many grammatical and typographical errors and to keep the conclusion part brief. 

Line 21: please revise as the most, Worldwide, Known

RE: Corrections implemented Line 21

Add comma before and in the following lines 22, 27, 51, 59, 91, 170, 767

RE: Corrections implemented Lines 22, 27, 51, 91, 175, 775

Line 23:  revise as Insight into

RE: Correction implemented Line 23

Line 25: remove letter (a) before winter

RE: Correction implemented Line 25

Line 54: revise as oxidase-positive

RE: Correction implemented Line 54

Line 82: revise as have been

RE: Correction implemented Line 82

Line 85: revise as isolated from

RE: Correction implemented Line 85

Line 87: revise as  describe

RE: Correction implemented Line 85

Line 165: the authors are encouraged to add the following information provided in the rebuttal letter to prove that the fish is Specific pathogen free.

RE: The information was added to Lines 158-165 as follows:

The fish facility and the fish are inspected by the provincial authorities and tested for reportable diseases and non-reportable diseases, including V. anguillarum. Prescence of these pathogens will cause the elimination of all the cohort. In addition, the seawater is intake at 30 mt deep, sand-filtered, UV treated, and degassed. The fish are produced from healthy cultivated broodstocks by in vitro fertilization under aseptic conditions. These measurements allow to have a facility specific pathogens free lumpfish. Also, in our experiments all non-infected animals are healthy and free of V. anguillarum, indicating that they are SPF lumpfish.

- regarding the feeding frequency: 3 times per day statement was not included, please add

RE: The information was added to Lines 170-171 as follows:

Fish were fed with a commercial feed (Skeretting-Europa; 55% crude protein, 15% crude fat, 3% calcium, 2% phosphorus, 1.5% crude fiber, 1% sodium, 5000 IU/kg vita-min A, 3000 IU/kg vitamin D, 200 IU/kg vitamin E), administered at a rate of 0.5% of body weight/day, 3 times per day.

- Line 168: n= should be in italic form

RE: Correction implemented Line 176

- Line 171: revise as described and the mid logarithmic phase

RE: Correction implemented Line 179

- Line 180: remove comma after ms222 and add comma before and spleen

RE: Correction implemented Line 188

- Line 181: revise as the weight

RE: Correction implemented Line 189

- Line 720: unneeded space 

RE: Correction implemented Lines 723-726

- Line 738: revise as was done

RE: Correction implemented Line 746

- Line 744: revise as with an N-formyl-L-alanyl group

RE: Correction implemented Line 752

Line 750: revise as the absence

RE: Correction implemented Line 758

Line 760: revise as the discovery

RE: Correction implemented Line 768

Authors are advised to concise the conclusion part, as it contains many sentences related to the result section. They should present only the important results and maximize their importance for the future

RE: The conclusion was revised as follows:

  1. anguillarum O1 J382, isolated from a natural infection in winter Steelhead trout in the Pacific coast and subjected to phenotypic and genomic characterization. V. anguillarum J382 isolate was highly virulent to Atlantic lumpfish, suggesting a broad host-fish range. Also, J383 was colistin sulphate resistant, like other V. anguillarum O1 strains. Comparative genomics of the O-antigen gene clusters of serotypes O1, O2 and O3 revealed genetic variation in genes related to nucleotide-sugar translocation mechanisms, O-antigen polymerization and composition. Also, gene rearrangements between serotypes were observed. ANI, phylogeny and synteny revealed the closest relatives of V. anguillarum J382 as well as genetic diversity among the different serotypes showing intra-species evolution. Some of the O-antigen biosynthesis pathways identified were earlier identified in distantly related Gram-negative bacteria such as P. aeruginosa indicating inter-species lateral transfer of the O-antigen gene clusters. ncRNAs were also annotated and they provided insight into the various putative functions these might carry out in V. anguillarum under physiological adaptation. This study contributes with essential pathogenesis and comparative genomics data of virulent V. anguillarum serotypes, critical for further fundamental studies (e.g., gene function) and application in fish health (e.g., vaccines).